# The *HDAC9*-associated risk locus promotes coronary artery disease by governing *TWIST1*

**Lijiang Ma**[1,2], **Nicole S. Bryce**[3], **Adam W. Turner**[4], **Antonio F. Di Narzo**[1], **Karishma Rahman**[2], **Yang Xu**[2], **Raili Ermel**[5], **Katyayani Sukhavasi**[5], **Valentina d'Escamard**[2], **Nirupama Chandel**[2], **Bhargavi V'Gangula**[2], **Kathryn Wolhuter**[3], **Daniella Kadian-Dodov**[6], **Oscar Franzen**[7], **Arno Ruusalepp**[5], **Ke Hao**[1,8], **Clint L. Miller**[4], **Johan L. M. Björkegren**[1,7], **Jason C. Kovacic**[2,3,6] *

1 Department of Genetics and Genomic Sciences, Icahn School of Medicine at Mount Sinai, New York, New York, United States of America, 2 Cardiovascular Research Institute, Icahn School of Medicine at Mount Sinai, New York, New York, United States of America, 3 Victor Chang Cardiac Research Institute, Darlinghurst, Australia; St Vincent's Clinical School, University of NSW, Sydney, Australia, 4 Center for Public Health Genomics, Department of Public Health Sciences, University of Virginia School of Medicine, Charlottesville, Virginia, Unites States of America, 5 Department of Cardiac Surgery and The Heart Clinic, Tartu University Hospital, Tartu, Estonia, 6 Zena and Michael A. Wiener Cardiovascular Institute and Marie-Josée and Henry R, Kravis Center for Cardiovascular Health Icahn School of Medicine at Mount Sinai, New York, New York, Unites States of America, 7 Integrated Cardio Metabolic Centre, Department of Medicine, Karolinska Institutet, Karolinska Universitetssjukhuset, Huddinge, Sweden, 8 Department of Respiratory Medicine, Shanghai Tenth People's Hospital, Tongji University, Shanghai, China

* j.kovacic@victorchang.edu.au

**Data Availability Statement:** All raw and processed single-cell chromatin accessibility sequencing datasets are made available on the Gene Expression Omnibus (GEO) database

## Abstract

Genome wide association studies (GWAS) have identified thousands of single nucleotide polymorphisms (SNPs) associated with the risk of common disorders. However, since the large majority of these risk SNPs reside outside gene-coding regions, GWAS generally provide no information about causal mechanisms regarding the specific gene(s) that are affected or the tissue(s) in which these candidate gene(s) exert their effect. The 'gold standard' method for understanding causal genes and their mechanisms of action are laborious basic science studies often involving sophisticated knockin or knockout mouse lines, however, these types of studies are impractical as a high-throughput means to understand the many risk variants that cause complex diseases like coronary artery disease (CAD). As a solution, we developed a streamlined, data-driven informatics pipeline to gain mechanistic insights on complex genetic loci. The pipeline begins by understanding the SNPs in a given locus in terms of their relative location and linkage disequilibrium relationships, and then identifies nearby expression quantitative trait loci (eQTLs) to determine their relative independence and the likely tissues that mediate their disease-causal effects. The pipeline then seeks to understand associations with other disease-relevant genes, disease sub-phenotypes, potential causality (Mendelian randomization), and the regulatory and functional involvement of these genes in gene regulatory co-expression networks (GRNs). Here, we applied this pipeline to understand a cluster of SNPs associated with CAD within and immediately adjacent to the gene encoding *HDAC9*. Our pipeline demonstrated, and validated, that this locus is causal for CAD by modulation of *TWIST1* expression levels in the arterial wall, and by also governing a GRN related to metabolic function in skeletal muscle. Our

(Accession GSE175621). The STARNET data is accessible through Database of Genotypes and Phenotypes (dbGAP), accession phs001203.v1.p1. A STARNET online resource is also available at starnet.mssm.edu.

**Funding:** KH acknowledges support from NIH (1R01ES029212). CLM acknowledges support from NIH (R01HL148239, R00HL125912) and Fondation Leducq. JLMB acknowledges support from NIH R01HL125863, Swedish Research Council (2018-02529) and Heart Lung Foundation (20170265), Foundation Leducq (PlaqOmics, 18CVD02; and CADgenomics, 12CVD02) and Astra-Zeneca. DKD acknowledges support from NIH (R01HL148167). JCK acknowledges support from NIH (R01HL130423, R01HL135093, R01HL148167), New South Wales health grant RG194194, the Bourne Foundation and Agilent. AWT acknowledges support from American Heart Association (20POST35120545). The funders had no role in study design, data collection and analysis, decision to publish, or preparation of the manuscript.

**Competing interests:** I have read the journal's policy and the authors of this manuscript have the following competing interests: JB and AR are shareholders in Clinical Gene Network AB that has an invested interest in STARNET. JK is the recipient of an Agilent Thought Leader Award (January 2022), which includes funding for research that is unrelated to the current manuscript. The remaining authors have nothing to disclose.

results reconciled numerous prior studies, and also provided clear evidence that this locus does not govern HDAC9 expression, structure or function. This pipeline should be considered as a powerful and efficient way to understand GWAS risk loci in a manner that better reflects the highly complex nature of genetic risk associated with common disorders.

## Author summary

Genome wide association studies (GWAS) have identified thousands of single nucleotide polymorphisms (SNPs) associated with the risk of common disorders. However, for the great majority of these SNPs, the causal mechanisms regarding the specific gene(s) that are affected or the tissue(s) in which these candidate gene(s) exert their effect are unknown. As a solution, we developed a streamlined, data-driven informatics pipeline to gain mechanistic insights on complex genetic loci. Here, we applied this pipeline to understand a cluster of SNPs associated with coronary artery disease (CAD) within and immediately adjacent to the gene encoding *HDAC9*. Our pipeline demonstrated, and validated, that this locus is causal for CAD by modulation of *TWIST1* gene expression levels in the arterial wall, and by also governing a gene regulatory co-expression network related to metabolic function in skeletal muscle. Our results reconciled numerous prior studies, and also demonstrated that this locus does not govern HDAC9 expression, structure or function. This pipeline should be considered as a powerful and efficient way to understand GWAS risk loci.

## Introduction

Coronary artery disease (CAD) is a complex disease driven by interactions between genetic and environmental factors [1,2]. In the last decade, genome wide association studies (GWAS) have identified thousands of variants that are associated with the risk of CAD and other common complex disorders [3–5]. While the GWAS approach has been remarkably successful, many questions have arisen [1]. In particular, because most identified risk loci are in non-coding genomic regions, GWAS provide little or no information about causal mechanisms. Indeed, GWAS do not provide any intrinsic information about the specific gene(s) that might be related to differing variants and how these genes might cause CAD [1]. At present, the 'gold standard' method for determining causal genes and their mechanisms of action are expensive and laborious basic science studies that often involve the creation of sophisticated knockin or knockout mouse lines [1]. However, the time-consuming nature of these studies, and their significant cost, means that they are totally impractical as a high-throughput means to understand the many risk variants that cause complex diseases like CAD [1]. Moreover, the repeated failure of the findings from small animal models of clinical diseases to be replicated or translated to humans [6] underscores the critical need to study and understand human genetics using humans as the model system.

As a solution, we recently developed a streamlined, data-driven informatics pipeline to gain tractable but reliable insights on the causal mechanisms of complex CAD genetic loci using exclusively human datasets [7]. Our pipeline leverages the STARNET study that comprises RNA sequence data from multiple tissues that were collected from living patients with advanced CAD and controls without CAD [2,7–10]. However, our pipeline is also applicable to other genetics-of-gene expression datasets such as GTEx [11]. We recently successfully

applied this pipeline to understand a cluster of single nucleotide polymorphisms (SNPs) in the vicinity of the coding region for Zinc Finger E-Box Binding Homeobox 2 (*ZEB2*) that are associated with risk of CAD. This revealed significant complexity and interaction among these SNPs, with independent disease-promoting effects of *ZEB2* in at least two separate CAD-relevant tissues, as well as a key driver role whereby *ZEB2* governs an important gene regulatory co-expression network (GRN) that is related to CAD [7].

As another critical genetic locus for CAD, there is major scientific controversy regarding a cluster of SNPs within and immediately adjacent to the 3' end of the gene encoding Histone deacetylase 9 (*HDAC9*) (Fig 1 and Table 1) [12,13]. In addition to CAD, variants in this locus are associated with large artery stroke [14,15] and atherosclerotic aortic calcification [16]. Because many of the SNPs in this locus physically reside within the *HDAC9* gene (Fig 1B), it has been widely assumed that their effects are mediated through *HDAC9* [5,17]. This is supported by several molecular studies that have documented a mechanistic role for HDAC9 as being causal for atherosclerosis [18,19] and vascular calcification [16]. Furthermore, Wang et al [12] reported that patients with CAD had higher levels of *HDAC9* mRNA expression and plasma HDAC9 than controls. Subsequent analyses indicated correlations of the lead SNP in this locus, rs2107595 (Fig 1 and Table 1), with *HDAC9* mRNA expression and plasma HDAC9 levels in controls and patients with myocardial infarction. This led the authors to conclude that rs2107595 is likely to contribute to atherosclerosis and CAD risk by regulating HDAC9 expression and gene-environment interactions [12]. Along similar lines, Azghandi et al reported increased mRNA levels of *HDAC9*, but not of neighboring genes, in risk allele carriers of rs2107595 in peripheral blood mononuclear cells, which combined with genetic knockout studies in mice led them to reach similar conclusions [19]. Finally, Prestel et al have also provided data to suggest that rs2107595 promotes atherosclerosis by controlling HDAC9 levels in inflammatory cells [20].

Despite this seemingly robust body of data, several studies have argued against *HDAC9* being the causal gene that is linked to this risk locus. These alternate studies do not dispel the notion that *HDAC9* is important for atherosclerosis and CAD, but rather, they suggest that this locus causes CAD by governing Twist-related protein 1 (*TWIST1*), rather than *HDAC9*. Notably, the gene encoding TWIST1 is also in the vicinity of the *HDAC9*-associated risk locus. These studies include Nurnberg et al [21] which showed that rs2107595 alters *TWIST1* expression in smooth muscle cells, and that *TWIST1* expression influences vascular smooth muscle cell phenotypes, including proliferation and calcification, as a potential mechanism supporting a role for *TWIST1* in CAD and atherosclerosis. This is supported by developmental studies characterizing transcriptional enhancers that regulate spatiotemporal *Twist1* activity, which identified 8 active enhancer candidates located in the region of *HDAC9-TWIST1* and which recapitulated aspects of the *Twist1* expression pattern during development [22]. In addition, both a large-scale mendelian randomization analysis [23] and an integrative genomics analysis [24] suggested that *TWIST1* is potentially the causal CAD gene related to this locus, and that this causal effect of *TWIST1* is operative in the arterial wall.

Given this surprising degree of disagreement in the literature, we felt there was an important need to apply our data-driven informatics pipeline [7] in order to understand this *HDAC9*-associated risk locus and how it orchestrates its disease-causal effects for CAD.

## Methods

### Ethics statement

Subject recruitment and tissue collection in the Stockholm-Tartu Atherosclerosis Reverse Networks Engineering Task study (STARNET) were performed as previously described [2,7–10].

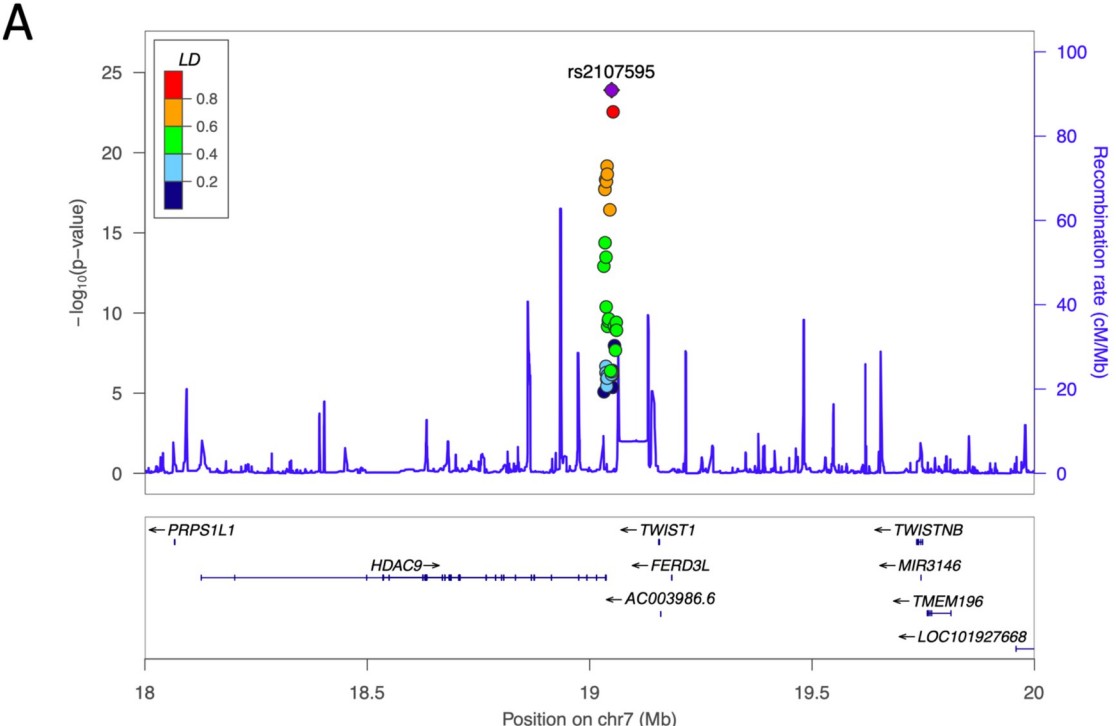

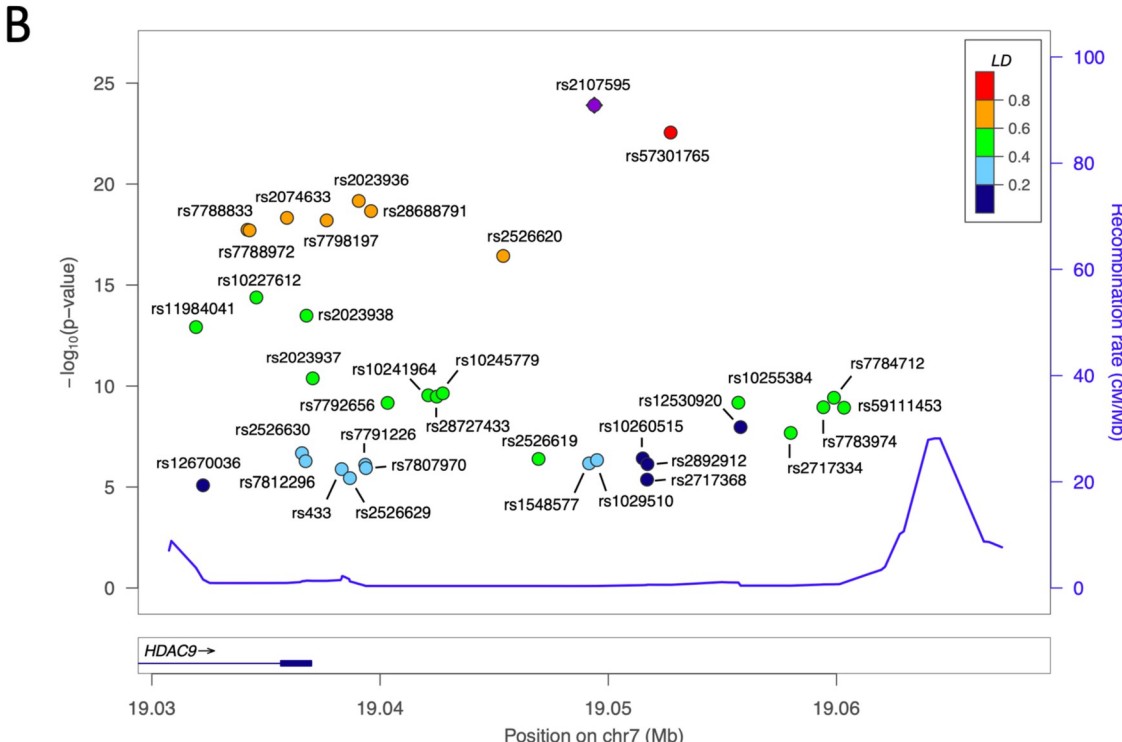

**Fig 1. GWAS SNPs and LD relationships in the *HDAC9*-associated CAD risk locus. (A)** Locus zoom plot of known GWAS SNPs related to the risk of CAD, stroke and atherosclerotic aortic calcification in the vicinity of *HDAC9* with rs2107595 showing the strongest correlation to CAD risk (lead SNP–shown in purple). Linkage disequilibrium (LD) relationships of each SNP with the lead SNP are color coded as indicated. **(B)** Further zooming of the locus plot as presented in A. LD relationships of each SNP with the lead SNP are color coded as indicated.

**Table 1. SNPs associated with CAD, stroke, atherosclerotic aortic calcification (AAC) in the *HDAC9*-associated risk locus and its vicinity.** SNPs correspond to those presented in Fig 1 and are ranked by P value. SNPs were obtained from published reports [13–16] and public databases, including the GWAS catalog [25] and PhenoScanner [26]. None of the SNPs had a stronger beta and/or smaller P value for AAC or stroke than for CAD, and therefore the beta, standard error and P value listed in Table 1 are for CAD.

| Rs ID | Chromosome Position | a1 | a2 | Beta | Standard Error | P value | Trait | References |
|---|---|---|---|---|---|---|---|---|
| rs2107595 | 7:19049388 | A | G | 0.0752 | 0.0073 | $1.25 \times 10^{-24}$ | CAD, stroke, AAC | 13,14,16 |
| rs57301765 | 7:19052733 | A | G | 0.0722 | 0.0073 | $2.81 \times 10^{-23}$ | CAD | 13 |
| rs2023936 | 7:19039067 | C | G | -0.0624 | 0.0068 | $6.84 \times 10^{-20}$ | CAD, AAC | 13,16 |
| rs28688791 | 7:19039605 | T | C | -0.0614 | 0.0068 | $2.18 \times 10^{-19}$ | CAD, AAC | 13,16 |
| rs2074633 | 7:19035920 | T | C | -0.0599 | 0.0067 | $4.70 \times 10^{-19}$ | CAD | 13 |
| rs7798197 | 7:19037661 | A | G | -0.0609 | 0.0068 | $6.29 \times 10^{-19}$ | CAD, AAC | 13,16 |
| rs7788833 | 7:19034191 | T | C | -0.0586 | 0.0067 | $1.83 \times 10^{-18}$ | CAD | 13 |
| rs7788972 | 7:19034280 | A | T | 0.0586 | 0.0067 | $1.96 \times 10^{-18}$ | CAD | 13 |
| rs2526620 | 7:19045397 | A | G | -0.0579 | 0.0069 | $3.63 \times 10^{-17}$ | CAD, AAC | 13,16 |
| rs10227612 | 7:19034579 | T | G | -0.0493 | 0.0063 | $4.11 \times 10^{-15}$ | CAD | 13 |
| rs2023938 | 7:19036775 | T | C | -0.0638 | 0.0084 | $3.27 \times 10^{-14}$ | CAD | 13 |
| rs11984041 | 7:19031935 | T | C | 0.0632 | 0.0085 | $1.20 \times 10^{-13}$ | CAD, stroke | 13,15 |
| rs2023937 | 7:19037051 | A | T | -0.0616 | 0.0093 | $4.15 \times 10^{-11}$ | CAD | 13 |
| rs10245779 | 7:19042749 | C | G | 0.0588 | 0.0093 | $2.30 \times 10^{-10}$ | CAD | 13 |
| rs10241964 | 7:19042114 | A | G | 0.0585 | 0.0093 | $2.87 \times 10^{-10}$ | CAD | 13 |
| rs28727433 | 7:19042484 | A | G | 0.0583 | 0.0093 | $3.34 \times 10^{-10}$ | CAD | 13 |
| rs7784712 | 7:19059890 | T | C | 0.0615 | 0.0098 | $3.79 \times 10^{-10}$ | CAD | 13 |
| rs10255384 | 7:19055703 | A | C | 0.0601 | 0.0097 | $6.68 \times 10^{-10}$ | CAD | 13 |
| rs7792656 | 7:19040331 | T | C | -0.0558 | 0.009 | $6.82 \times 10^{-10}$ | CAD | 13 |
| rs7783974 | 7:19059427 | T | C | 0.0594 | 0.0098 | $1.12 \times 10^{-9}$ | CAD | 13 |
| rs59111453 | 7:19060333 | C | G | -0.0596 | 0.0098 | $1.18 \times 10^{-9}$ | CAD | 13 |
| rs12530920 | 7:19055797 | A | C | -0.0543 | 0.0095 | $1.08 \times 10^{-8}$ | CAD | 13 |
| rs2717334 | 7:19057996 | C | G | 0.051 | 0.0091 | $2.10 \times 10^{-8}$ | CAD | 13 |
| rs2526630 | 7:19036578 | T | C | 0.0295 | 0.0057 | $2.11 \times 10^{-7}$ | CAD | 13 |
| rs10260515 | 7:19051512 | C | G | -0.0288 | 0.0057 | $3.81 \times 10^{-7}$ | CAD | 13 |
| rs2526619 | 7:19046946 | A | G | -0.0454 | 0.009 | $4.07 \times 10^{-7}$ | CAD | 13 |
| rs1029510 | 7:19049507 | A | T | 0.0288 | 0.0057 | $4.64 \times 10^{-7}$ | CAD | 13 |
| rs7812296 | 7:19036738 | T | C | -0.0285 | 0.0057 | $5.24 \times 10^{-7}$ | CAD | 13 |
| rs1548577 | 7:19049162 | A | G | 0.0283 | 0.0057 | $6.76 \times 10^{-7}$ | CAD | 13 |
| rs2892912 | 7:19051715 | A | T | -0.028 | 0.0057 | $7.33 \times 10^{-7}$ | CAD | 13 |
| rs7791226 | 7:19039351 | T | C | 0.028 | 0.0057 | $7.87 \times 10^{-7}$ | CAD | 13 |
| rs7807970 | 7:19039381 | C | G | 0.0276 | 0.0057 | $1.16 \times 10^{-6}$ | CAD | 13 |
| rs433 | 7:19038319 | A | G | 0.0275 | 0.0057 | $1.30 \times 10^{-6}$ | CAD | 13 |
| rs2526629 | 7:19038678 | A | G | -0.0263 | 0.0057 | $3.62 \times 10^{-6}$ | CAD | 13 |
| rs2717368 | 7:19051700 | T | C | -0.0262 | 0.0057 | $4.30 \times 10^{-6}$ | CAD | 13 |
| rs12670036 | 7:19032243 | A | G | -0.0351 | 0.0079 | $8.21 \times 10^{-6}$ | CAD | 13 |

Briefly, patients with angiographically proven CAD who were eligible for open-thorax surgery and control subjects without CAD were enrolled into this institutional review committee approved protocol after written informed consent at Tartu University Hospital, Estonia.

## STARNET study description, tissue collection and processing

From each STARNET subject, venous blood (BLOOD) as well as biopsies from atherosclerotic aortic wall (AOR), pre/early-atherosclerotic mammary artery (MAM), liver (LIV), skeletal

muscle (SKLM), subcutaneous fat (SF) and visceral fat (VAF) were obtained and RNA was extracted. BLOOD was also used to obtain macrophages (MP) and foam cells (FC), as well as DNA. RNA sequencing (RNA-seq) of STARNET samples and processing of DNA was performed as described [7]. The STARNET data is accessible through Database of Genotypes and Phenotypes (dbGAP), accession phs001203.v1.p1.

## Previously identified SNPs associated with CAD and other CVDs in the *HDAC9* locus

Known GWAS SNPs associated with CAD, ischemic stroke, aortic calcification and other CVDs in the *HDAC9*-associated risk locus and its vicinity were obtained from published reports [13–16] and public databases, including the GWAS catalog [25] and PhenoScanner [26]. These GWAS SNPs were plotted and visualized in LocusZoom [27]. Linkage disequilibrium (LD) scores of the GWAS SNPs were calculated using 1000 Genome data with all ancestry groups, including African American, Hispanic, European and Asian, by LDlinkR in R (4.0.2) [28].

## Causal variant fine-mapping in credible set analysis

CAUSALdb GWAS fine-mapping tool was used for causal variant fine-mapping credible set analysis [29]. CAD GWAS summary statistics were obtained from van der Harst P et al [13]. LD blocks were estimated using LDetect [30]. Fine-mapping based on summary statistics and matched LD matrix for each causal block of each trait was performed using FINEMAP [31].

## Genomic mapping of SNPs with relation to the *HDAC9* gene

The genomic DNA sequence of human *HDAC9* and 50 Kb downstream sequence (GRCh38/hg38) was downloaded from http://genome.ucsc.edu [32]. SNP flanking sequences were identified using https://www.ncbi.nlm.nih.gov/snp/ and mapped in SnapGene Version 5.3.2. Predicted 3'UTR sequence motifs were identified using UTRScan [33].

## STARNET RNA sequencing data and quality control

RNA-seq quality control was performed using FASTQC [34] that checks raw sequence data for per-base quality, per-sequence quality, number of duplicate reads, number of reads with an adaptor, sequence length distribution, per-base GC content, per-sequence GC content and Kmer content. GENCODE v.19 was used as reference annotation to quantify gene and isoform expression. Sequencing reads (fastq files) were mapped with STAR [35] onto the human genome. Raw reads were summarized by feature counts [36]. Samples with less than 1,000,000 uniquely mapped reads were discarded. Low counts were removed by keeping genes where the count per million (cpm) is greater than 1 in at least two samples.

## STARNET SNP-based genotyping, imputation and expression quantitative trait loci analysis

DNA SNP-based genotyping was performed using the Illumina Infinium assay with the human OmniExpressExome-8v1 bead chip. Data was analyzed using GenomeStudio 2011.1 (Illumina) which produced 951,117 genomic markers (genome build 37) [8]. Quality control was performed using PLINK v.1.07, and IMPUTE2 v.2.3.0 was used for genotype imputation to increase the power of analysis [8]. In each tissue, cis-regulated expression quantitative trait loci (eQTLs) were identified with the R package Matrix eQTL v.2.1.1 [37]. Per convention, we defined a cis-eQTL as residing within 1 MB upstream or downstream of the *HDAC9*-

associated risk locus. Adjusted FDR < 0.05 was considered significant for eQTL analyses. After the above steps, the following numbers of subjects were available for the final eQTL analysis for each tissue: AOR 515, MAM 520, LIV 523, SKLM 514, BLOOD 471, SF 550, VAF 509, FC 81, MP 96.

## Transcriptomic data from the Genotype-Tissue Expression (GTEx) project

*TWIST1*, *HDAC9*, *FERD3L* and AC003986.6 gene expression in GTEx RNA-seq data was downloaded from GTEx V8 release [11]. Although GTEx contained 48 tissues in its datasets, many of these tissues are unlikely to be related to CAD (e.g. uterus, bladder, esophagus, tibial nerve). Therefore, we only considered the following GTEx tissues that have biologic plausibility for causing CAD: SF, VF, AOR, LIV, SKLM, BLOOD and also coronary artery (COR) and Tibial Artery (TA). Note that while GTEx allowed us to analyze SF, VF, AOR, LIV, SKLM and BLOOD that were all in STARNET, and also to analyze COR and TA that were not in STARNET, on the other hand GTEx does not have MAM, MP or FC and therefore these tissues/cell types were excluded from GTEx validation analyses. Cis-regulated eQTLs in GTEx for CAD GWAS SNPs were downloaded from the open access GTEx portal.

## Association of *TWIST1* expression levels and clinical phenotypes

STARNET phenotypic and demographic data were used to explore relationships with *TWIST1* and *HDAC9* expression levels. These data included indices of CAD severity: Duke CAD index [38], SYNTAX score [39], number of coronary lesions per patient (Lesions) and number of diseased coronary vessels per patient (Diseased vessels). CAD DGE represents the enrichment of differential gene expression in the module between cases and controls. Other data included systolic and diastolic blood pressure, body mass index (BMI) and waist to hip ratio (Waist: Hip); and the following parameters from peripheral blood: c-reactive protein concentration (CRP), hemoglobin A1c concentration (HbA1c), hemoglobin concentration (HbG), creatinine level, fasting plasma cholesterol concentration (P-Chol), fasting plasma low-density lipoprotein cholesterol concentration (fP-LDL-Chol), fasting plasma high-density lipoprotein cholesterol concentration (fP-HDL-Chol), fasting plasma triglyceride concentration (fP-TG). Pearson correlation coefficient in R (4.0.2) was used to measure the strength of linear correlation between expression of *TWIST1* in AOR and clinical phenotypes.

## Co-expression module, network and key driver analysis

Detailed methods are provided in our prior publication [10]. In brief, R package Weighted Gene Co-expression Network Analysis (WGCNA) [40] was used to identify correlation patterns among genes across RNA-seq data. Gene regulatory co-expression networks (GRNs) were constructed based upon 30,716 genes from seven tissues. Unsigned, weighted correlation network construction and module detection were performed using default parameters. Genes with similar expression patterns were assigned into the same module. The correlations of differing modules with clinical traits were assessed using Pearson's correlation. Regulatory networks were reconstructed using GENIE3, an algorithm inferring gene regulatory networks from expression data based on feature selection with tree-based ensemble methods, by using transcription factor and eQTL regulated genes as regulators [41]. Key driver analyses were performed using Mergeomics [42]. Note that due to limitations of GENIE3, for large networks (greater than ~3000 genes), it is not possible to infer key drivers [10].

## Two sample, gene-level Mendelian Randomization

Within each STARNET tissue (AOR, BLOOD, LIV, MAM, SKLM, SF, VAF, FC, MP), we tested for causal association with CAD using a 2-sample Mendelian Randomization method [43]. Phenotype (outcome) and mRNA (exposure) association statistics were then combined using the inverse-variance weighted method, as implemented in the MendelianRandomization R package (version 0.4.3) [44]. In terms of effect estimates for this analysis, cis-eQTL data from STARNET was used as per S1 Table (effects of rs2107595 on *TWIST1* expression in AOR and MAM), while CAD association statistics were obtained from Nelson et al [3] for effects of rs2107595 on CAD risk (variant ID 7:19049388:G:A; effect allele frequency 0.18163; estimate 0.07422; standard error 0.01019; P value $3.41 \times 10^{-13}$; sample size 336847).

## Human coronary artery single cell ATAC-seq sample processing

Single cell ATAC-seq (scATAC-seq) of human coronary artery samples was performed for 41 patients using the 10x Genomics Single Cell ATAC platform as described previously [45]. These coronary arteries were obtained from explanted hearts from transplant recipients and hearts that were rejected for orthotopic heart transplantation (Stanford University School of Medicine). All samples were de-identified, and subjects provided consent through IRB-approved protocols for research study participation. Proximal coronary artery segments from either the left anterior descending artery, left circumflex artery, or right coronary artery were placed in cardioplegic solution before dissection. After dissection, samples were flash frozen in liquid nitrogen and stored at -80˚C.

For the scATAC-seq experiments, all coronary samples were broken into small fragments using a chilled mortar and pestle with dry ice and liquid nitrogen. Nuclei were then isolated using a Iodixanol/sucrose gradient and density gradient centrifugation as per the Omni ATAC-seq protocol [46]. After isolation, the nuclei were first transposed with Tn5 transposase in bulk and gel beads in emulsions (GEMs) were then captured using the 10x Genomics Chromium Controller instrument. Single cell ATAC-seq libraries were prepared using the 10x Genomics Chromium Single Cell ATAC Kit (version 1 or 1.1 (Next GEM)). Barcoded libraries were subsequently sequenced using the Illumina NovaSeq 6000 (Icahn School of Medicine at Mount Sinai, New York, NY).

## Single cell ATAC-seq analysis

Coronary artery single cell ATAC-seq data was preprocessed using the 10x Genomics Cell Ranger ATAC pipeline (version 1.2.0) and reads mapped to the hg38 reference genome. The downstream scATAC-seq analyses were performed using the ArchR software package (version 1.0.1) [47]. Using ArchR we filtered to keep high quality/informative nuclei (transcription start site enrichment $\geq 7$ and $\geq 10000$ unique fragments). To link chromatin accessibility with gene expression we integrated the scATAC-seq data with human coronary artery scRNA-seq data from 4 individuals [48]. Briefly, the scATAC-seq gene score (chromatin accessibility) matrix was compared with the scRNA-seq gene expression matrix in ArchR. Each cell in the scATAC-seq space was subsequently assigned the gene expression profile of the closest-matching cell from the coronary scRNA-seq. After integration, the "getPeak2GeneLinks" function in ArchR was used to correlate chromatin accessibility within peak windows with integrated RNA expression levels across all cell types. Finally, the bedtools package was used to intersect positions of CAD GWAS SNPs at the *HDAC9*-associated locus with genomic coordinates of scATAC peaks that correlate with integrated RNA expression (hg38). All raw and processed single-cell chromatin accessibility sequencing datasets are available on the Gene Expression Omnibus (GEO) database (Accession GSE175621).

## Results

We applied our informatics pipeline [7] to the *HDAC9*-associated CAD risk locus. The pipeline begins by understanding the SNPs in this locus in terms of their relative location and linkage disequilibrium (LD) relationships, and then uses gene expression data to identify eQTLs at these SNPs to further determine their relative independence and the likely tissues in which their CAD-causal effects are operative. The pipeline then seeks to understand additional associations with other CAD-relevant genes, CAD sub-phenotypes, potential causality, and also the involvement of these genes in key gene regulatory co-expression networks (GRNs) involved in CAD pathogenesis.

### SNPs in the *HDAC9*-associated CAD risk locus partially reside in the *HDAC9* coding region and are in linkage disequilibrium, but do not affect HDAC9 protein structure

Close examination of the GWAS SNPs in the *HDAC9*-associated CAD risk locus indicated that rs2107595 is the lead SNP (GWAS P value = $1.25 \times 10^{-24}$) [13,14,16], although rs57301765 is in high LD with rs2107595 ($r^2 = 0.96$ in Europeans) and has a similar GWAS P value ($P = 2.81 \times 10^{-23}$) (Fig 1A and 1B and Table 1) [13].

A significant proportion of SNPs in this cluster reside in the final intron and coding exon of *HDAC9* (Fig 1B and Table 1). While there are 3 other genes in close proximity to this locus (AC003986.6, *FERD3L*, *TWIST1*), none of these GWAS SNPs are located within the exons or introns of these other genes (Fig 1B and Table 1).

Consistent with the close physical clustering of the SNPs, there were strong LD relationships among the SNPs in this locus (Fig 2). Both based on the LD plots (Fig 2), and also the LD relationships to the lead SNP rs2107595 (Fig 1), it is evident that there are no sub-clusters of SNPs to suggest more than one potentially causal CAD SNP in this locus. Rather, these LD relationships were consistent with there being only a single causative SNP in this locus. We undertook a credible set analysis to further explore these relationships, which showed that there was 98.4% probability of this locus containing one true causal variant, with rs2107595 being the causal and lead variant for CAD in this region.

Given that many of the SNPs in the *HDAC9*-associated risk locus physically reside within the *HDAC9* gene (Fig 1), we considered if they might alter HDAC9 protein structure and function. Genomic mapping of human *HDAC9* in relation to the SNPs in this risk locus (Table 1) revealed that none of the SNPs map to a protein coding region of *HDAC9*. As shown in Fig 3, the SNPs in this risk locus were located either within the final intron, the 3'UTR, or the intergenic region of the *HDAC9* gene. Predicted motifs in the 3'UTR as detected by UTRScan [33], which comprised 3 UNR binding sites and 2 K-boxes, were not altered when either allele of the SNPs in this region were expressed.

### Tissue-specific expression levels of genes in the *HDAC9*-associated CAD risk locus

We next used STARNET to explore the tissue-specific expression levels of genes in close proximity to the CAD SNPs in this locus, being *HDAC9*, *FERD3L*, *TWIST1* and the non-coding gene AC003986.6 (Fig 1). Although the expression levels of *TWIST1* in SF were the highest of any gene in any tissue examined (~18 RPKM), the levels of *HDAC9* and *TWIST1* were otherwise broadly comparable (Fig 4A and 4B). In general, both *HDAC9* and *TWIST1* exhibited higher expression levels in AOR, MAM, MP, SF, and VAF (in the range of 2–18 RPKM). SKLM exhibited modest levels of *HDAC9* expression (~5 RPKM), but lower levels of *TWIST1*.

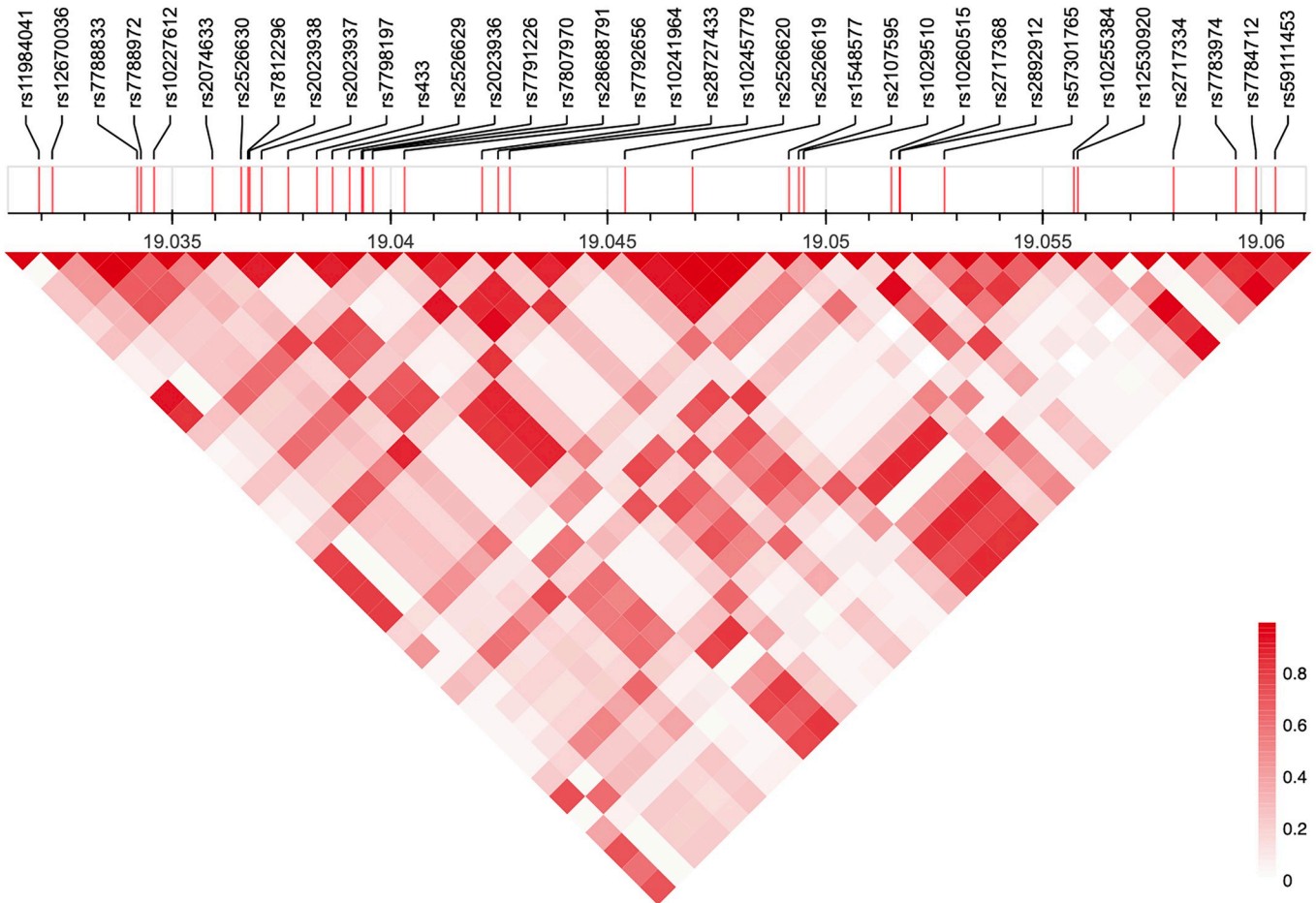

**Fig 2. LD relationships among GWAS SNPs in the *HDAC9*-associated CAD risk locus.** Shown here is the full LD heatmap for GWAS SNPs in the *HDAC9*-associated CAD risk locus.

Levels of AC003986.6 and *FERD3L* were notably less than for *HDAC9* and *TWIST1*, with the single exception being AC003986.6 in SF at ~3 RPKM (Fig 4A and 4B).

We also explored the levels of these transcripts in the GTEx project [11]. As shown in S1 Fig, both the patterns and transcript levels were similar between STARNET and GTEx, although in GTEx the levels of AC003986.6 were low in all tissues.

## SNPs in the *HDAC9*-associated CAD risk locus modulate expression levels of *TWIST1* but not *HDAC9* in the arterial wall

We sought to determine potential gene-regulatory effects for these CAD-associated SNPs in the *HDAC9*-associated risk locus by exploring their effect on transcript expression (i.e. eQTLs) in STARNET tissues. As shown in S1 Table, a majority of these GWAS SNPs were found to function as cis-eQTLs for the levels of *TWIST1* expression in the arterial wall. Of the 36 GWAS SNPs considered in this study (Table 1), 22/36 were eQTLs for *TWIST1* in AOR, and 34/36 were eQTLs for *TWIST1* in MAM. Importantly, in both AOR and MAM, the strongest eQTL associations for *TWIST1* were with the lead SNP rs2107595 (adjusted P value = 8.6 x $10^{-5}$ and 9.5 x $10^{-24}$, respectively for AOR and MAM).

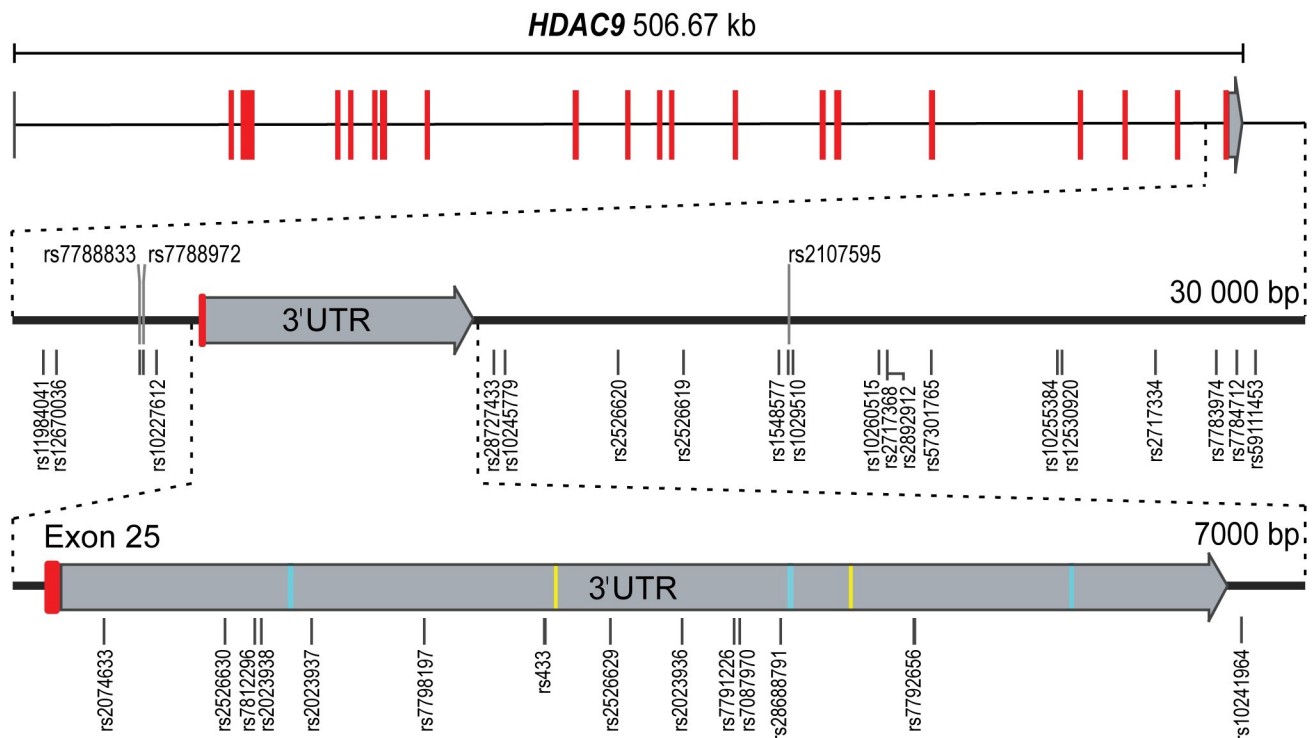

**Fig 3. Genomic location of GWAS SNPs in the *HDAC9*-associated CAD risk locus.** GWAS SNPs associated with the risk of CAD as shown in Table 1 are located within a 28,399 bp region on chromosome 7 encompassing the final intron and coding exon of *HDAC9* and the intergenic region between *HDAC9* and *TWIST1*. Red bars indicate protein coding exons, grey the 3'UTR of *HDAC9*. Predicted 3'UTR motifs are indicated by cyan bars (UNR binding site) and yellow bars (K-box). 3'UTR and SNP locations are drawn to scale, red-colored coding exons are enlarged for ease of viewing.

Apart from these multiple and relatively robust eQTLs for *TWIST1* in the arterial wall, there were 2 additional *TWIST1* eQTLs in VAF at non-lead SNPs, however, these were relatively weak (adjusted P value = 0.021 in both cases). There were also 4 eQTLs for AC003986.6 in MAM that included one at the lead SNP rs2107595, however, these were also relatively weak (adjusted P values ranged from 0.03–0.0005). There were no eQTLs for AC003986.6 in AOR or any other STARNET tissues. There were also no eQTLs for *HDAC9* or *FERD3L* in any STARNET tissues (S1 Table).

In GTEx, evaluation of these SNPs in the *HDAC9*-associated CAD risk locus corroborated the findings in STARNET and identified 10 eQTLs for *TWIST1* in AOR, with additional eQTLs also for AC003986.6 in AOR (S2 Table). Similar to the findings in STARNET, the strongest eQTLs were for *TWIST1*, again with rs2107595 being a robust eQTL for TWIST1 in AOR (P = 5.7 x $10^{-7}$), but with the strongest *TWIST1* eQTL being at rs57301765 (P = 3.2 x $10^{-7}$). Also, similar to STARNET, there were no eQTLs for *HDAC9* or *FERD3L* (S2 Table).

As shown in the eQTL violin plots for *TWIST1* in AOR for both STARNET (Fig 5) and GTEx (S2 Fig), the risk allele (on the right) is consistently associated with increased *TWIST1* expression levels. In contrast, the protective allele (on the left) is associated with reduced levels of *TWIST1*.

Collectively, these data indicate several points about the *HDAC9*-associated CAD risk locus: 1) The lead SNPs rs2107595 exhibited the strongest eQTLs in STARNET, adding to the likelihood that it is the true CAD-associated SNP in this locus; 2) The strongest eQTLs were for *TWIST1* in the arterial wall, with the risk alleles consistently being associated with increased *TWIST1* levels, suggesting that increased *TWIST1* levels in the arterial wall may be

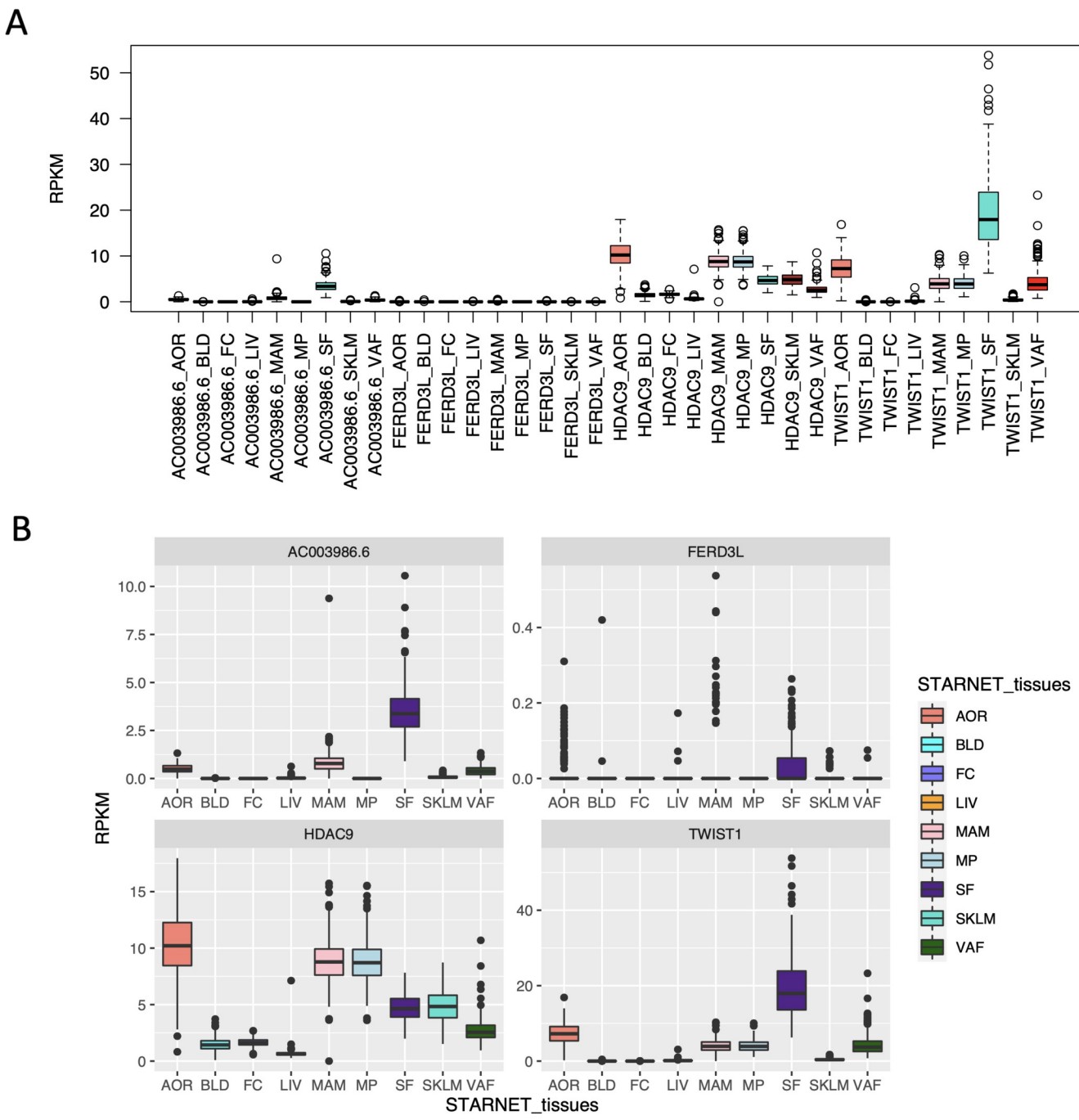

**Fig 4. Gene expression levels in STARNET. (A)** Gene expression levels of *HDAC9*, *TWIST1*, *FERD3L* and AC003986.6 in STARNET–which generally show higher expression levels of *HDAC9* and *TWIST1* across various tissue subtypes. **(B)** Same as A but at different scales for each gene. RPKM, Reads Per Kilobase of transcript per Million mapped reads.

causal for atherosclerosis and CAD; 3) Although prior studies suggested that this locus may mediate risk for atherosclerosis and CAD via HDAC9 signaling in inflammatory cells [19,20], there were no eQTLs for *HDAC9* in any tissues in either STARNET or GTEx. This is important considering that STARNET included both macrophages (MP) and foam cells (FC).

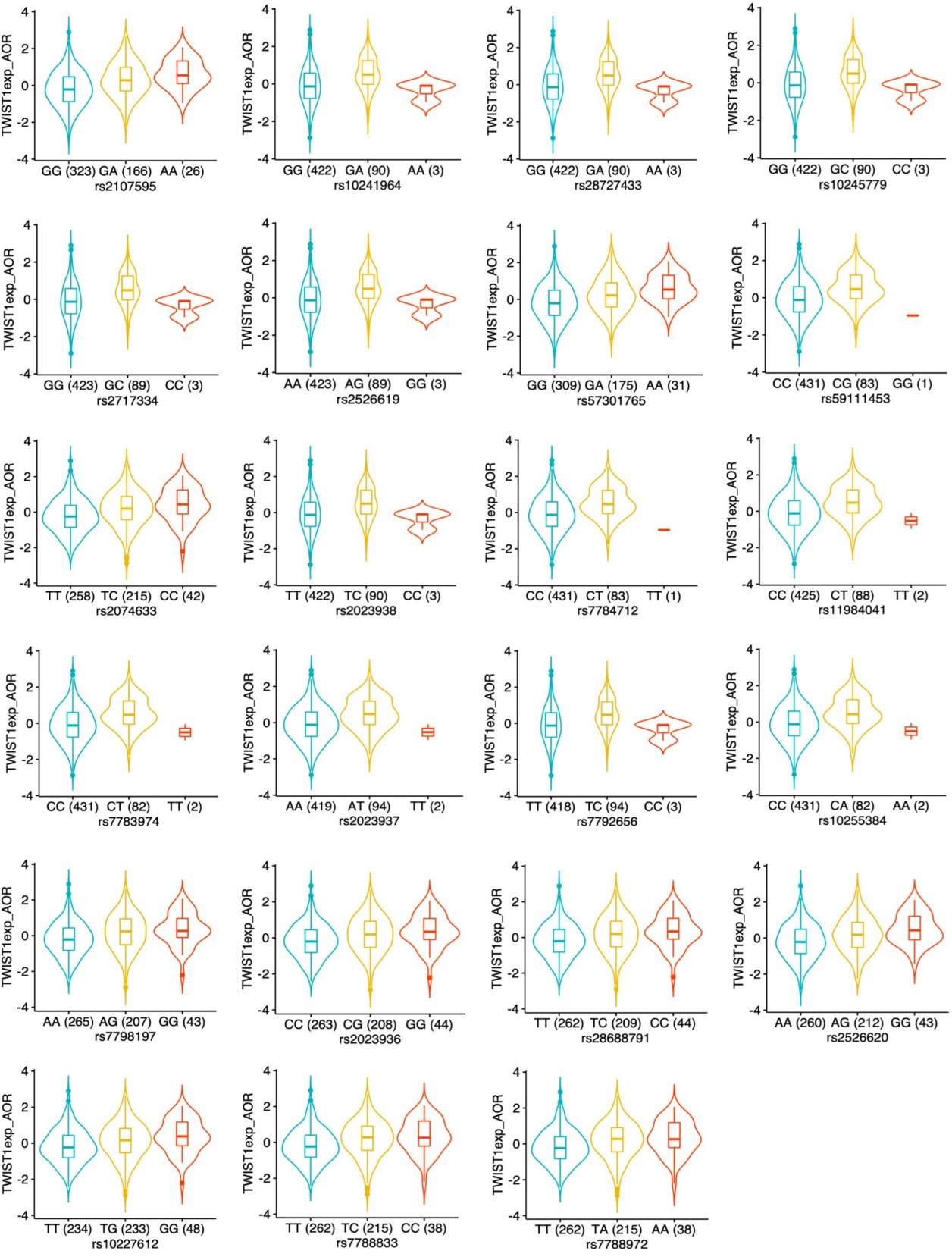

**Fig 5. Cis-eQTLs for *TWIST1* in the *HDAC9*-associated CAD risk locus in AOR in STARNET.** eQTL violin plots are shown for tissue expression of *TWIST1* in AOR in STARNET corresponding to SNPs in the *HDAC9*-associated CAD risk locus. The CAD risk allele is presented on the right in all panels, indicating that the CAD risk allele is consistently associated with higher expression levels of *TWIST1* in AOR. eQTLs are shown for FDR ≤ 0.05. This figure corresponds to S1 Table.

## *HDAC9*-associated CAD risk SNPs target *TWIST1* via accessible chromatin in coronary artery cell types

To explore whether these non-coding SNPs located near *HDAC9* could point to candidate genes through enhancer gene linkage, we leveraged our single-cell chromatin accessibility (scATAC-seq) atlas in human coronary artery (S3 Fig) [45]. From these data we observed that the lead CAD GWAS SNP rs2107595 (Table 1), which is located just 3' of *HDAC9*, resides in a candidate cis-regulatory element that links to the promoter of *TWIST1* (Fig 6). These accessible peaks in smooth muscle cells and fibroblasts are correlated with *TWIST1* RNA expression (R = 0.5665, FDR = 2.36 x $10^{-42}$), but not *HDAC9* or other genes in the locus, when using the integrated human coronary artery scATAC/scRNA data. Taken together, these results provide complementary support for our eQTL-based fine-mapping approaches in STARNET and confirm that the *HDAC9*-associated CAD risk locus operates through regulation of *TWIST1* in vascular cell types.

## *TWIST1* levels in the aortic wall are associated with burden of coronary artery disease

Because coronary arteries and AOR both develop atherosclerosis, whereas MAM typically does not, results using AOR appear to mirror findings in the coronary arteries. Furthermore, some of the risk SNPs in the *HDAC9*-associated locus are associated with atherosclerotic aortic calcification (Table 1). Therefore, we sought to identify phenotypic associations of *TWIST1* expression levels in AOR, and found that *TWIST1* levels were inversely associated with BMI and HbA1c levels (S3 Table). Furthermore, and potentially of greatest importance, *TWIST1* expression levels in AOR were also positively associated with the severity of CAD as assessed by Duke CAD index (P = 0.012), and with non-significant trends also for SYNTAX score (P = 0.12) and the number of diseased coronary vessels (P = 0.065) (S3 Table). The observation that greater levels of *TWIST1* in AOR are related to more severe CAD is consistent with the eQTL data where the CAD risk allele at rs2107595 and other SNPs in this locus exhibit higher levels of *TWIST1* expression in AOR (Figs 5 and S2). While not being direct proof, this consistency of findings in terms of the eQTLs for *TWIST1*, *TWIST1* expression levels in AOR and increased severity of CAD, collectively adds weight to the likelihood that the *HDAC9*-associated CAD risk locus exerts a CAD-causal effect by modulating *TWIST1* levels.

Similarly, we also evaluated the phenotypic associations of *HDAC9* expression levels in AOR and found a positive association with the severity of CAD as assessed by number of diseased coronary vessels (P = 0.021), and with a trend also for a positive association with Duke CAD index (P = 0.06) (S3 Table).

## Mendelian randomization analysis confirms that *TWIST1* is causative for CAD

Given the above, we undertook a Mendelian randomization (MR) analysis to further explore CAD causality at this locus using GWAS summary association statistics from Nelson et al [3] and cis-eQTL association statistics from the STARNET study [2,7–9].

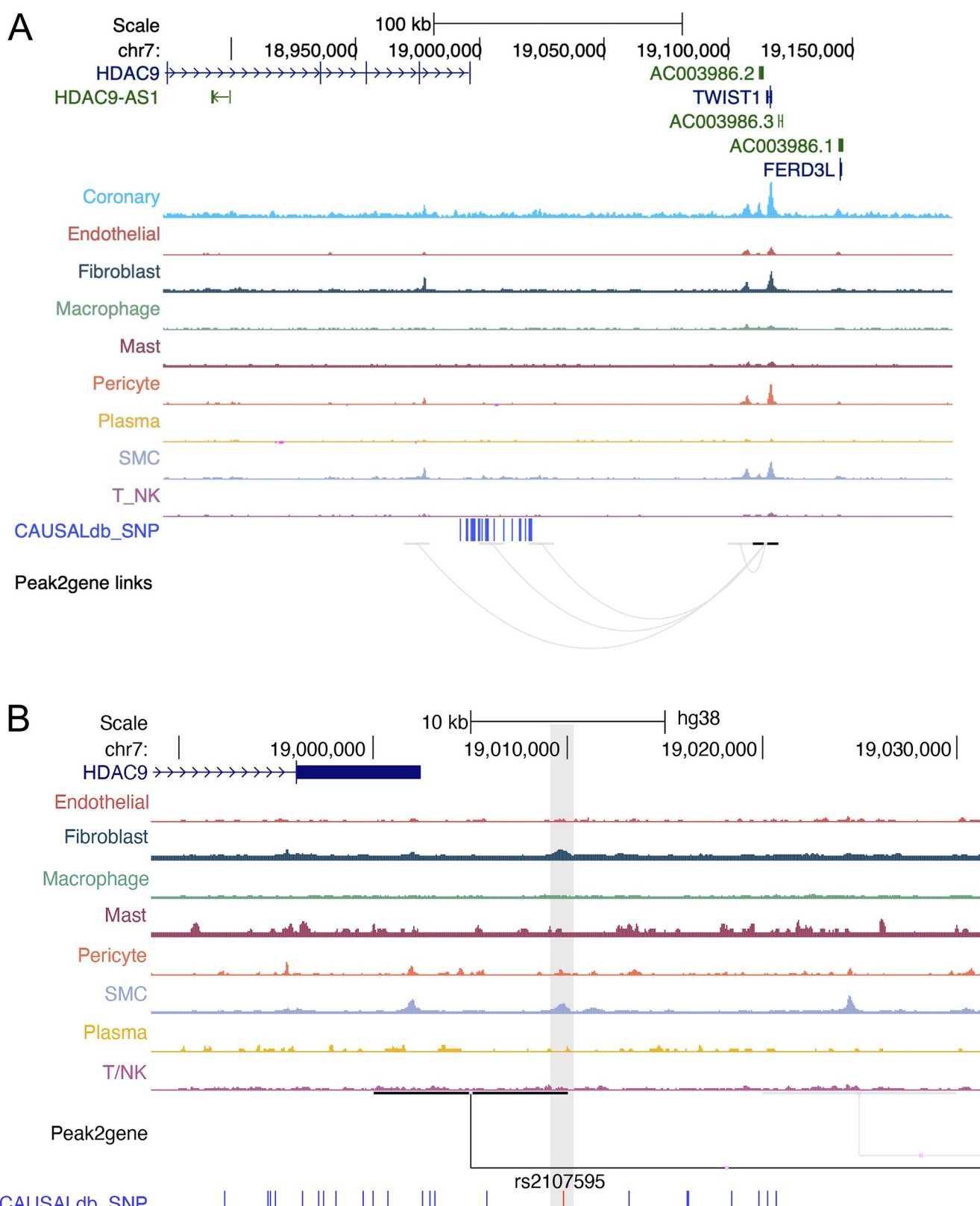

**Fig 6. rs2107595 links to *TWIST1* via cis-regulatory element in coronary artery cell types.** Genome browser showing single-cell chromatin accessibility (scATAC-seq) profiles in human coronary artery cell types shown **(A)** across the wider *HDAC9*-associated CAD risk locus (hg38) and **(B)** a zoomed in view

of the lead SNP rs2107595 highlighting open chromatin peaks and peak2gene links overlapping rs2107595 and top LD SNPs. Fine-mapped CAD GWAS SNPs from CAUSALdb are shown in blue. Peaks represent normalized accessibility as calculated in ArchR, with all tracks visualized in the UCSC browser at equivalent window sizes. The 'Coronary' track (uppermost track in A) represents a bulk ATAC-seq library from 50,000 nuclei (therefore consisting of several cell types) generated from human coronary artery [45]. Loops denote peak2gene based correlation between accessibility at gene regulatory elements (410 bp) and gene expression using the integrated human coronary artery scATAC-seq/scRNA-seq data for genes at the *HDAC9*-associated CAD risk locus.

The candidate causal variant rs2107595 had a significant cis-association with *TWIST1* expression levels in AOR (P = 6.8 x 10$^{-6}$) and MAM (P = 2.7 x 10$^{-22}$). Associations with *TWIST1* and *HDAC9* mRNAs in other STARNET tissues were all non-significant (all P > 0.01). Therefore, we used the ratio estimator from the Mendelian Randomization R package [44] to test for a significant causal effect of *TWIST1* mRNA in AOR and MAM. We also conducted the Heterogeneity In Dependent Instrument (HEIDI) test [49] to further exclude genes with evidence of heterogeneity in their cis-eQTLs. The HEIDI test indicated a significant heterogeneity of effects for MAM (P = 0.001, n = 20 SNPs) but not AOR (P = 0.1, n = 9 SNPs). While we cannot fully eliminate the possibility of a minor effect arising from horizontal pleiotropy, we finally estimated a positive causal effect of *TWIST1* in AOR (β = 1.87, standard error = 0.26, P = 0.00325), in that increased expression of *TWIST1* in AOR is associated with increased risk of CAD.

## Gene regulatory co-expression networks and key drivers

As described [10], gene regulatory co-expression networks (GRNs) were generated from AOR, MAM, BLOOD, LIV, SKLM, SF and VAF tissues in STARNET. A total of 224 GRNs were generated, with the majority of GRNs involving genes from multiple tissues [10]. *TWIST1* was found in 5 different GRNs, where it acted in AOR, MAM, LIV, SKLM, SF and VAF (Fig 7A). *TWIST1* was also identified as hierarchically being among the top key drivers for GRN 116 (P = 1.63 x 10$^{-32}$; Fig 7B). GRN 116 was predominantly located in SKLM and was associated with certain metabolic indices including BMI, Waist:Hip ratio and HbA1c (Fig 7C). Importantly, GRN 116 was also associated with the severity of CAD as assessed by various parameters including number of coronary lesions, number of diseased vessels and Duke CAD index (Fig 7C). Apart from GRN 116, *TWIST1* was not a key driver in any other GRNs.

*HDAC9* was found in 7 different GRNs, where it acted across multiple tissues (Fig 7D). Six of these 7 GRNs that included *HDAC9* were associated with the severity of CAD (S4 Fig). However, *HDAC9* was not a key driver in any of these networks. Other genes in the *HDAC9*-associated CAD risk locus were either not in any GRNs whatsoever (*FERD3L*) or were filtered out during construction of the GRNs due to very low gene expression levels (AC003986.6).

Collectively these GRN data, which are agnostic to the GWAS SNPs in the *HDAC9*-associated CAD risk locus, confirm that both *HDAC9* and *TWIST1* are likely involved in the pathogenesis of CAD, but also suggest an additional key driver role for *TWIST1* in GRN 116 in SKLM. Interestingly, similar to the phenotype associations identified for *TWIST1* expression levels in AOR (S3 Table), GRN 116 also appears to be related to the severity of CAD.

## Discussion

While GWAS have identified a wealth of disease-associated SNPs and gene loci, there still remains a considerable gap in resolving the causal mechanisms of these associations, which will be required to translate this knowledge back to patients [1]. Indeed, as this study has highlighted, in many cases we do not even know the potential causal gene that is linked to any given disease-associated SNP(s). As perhaps the most important message to emerge from this study, we have shown that our data-driven informatics pipeline [7] is an efficient, accurate and

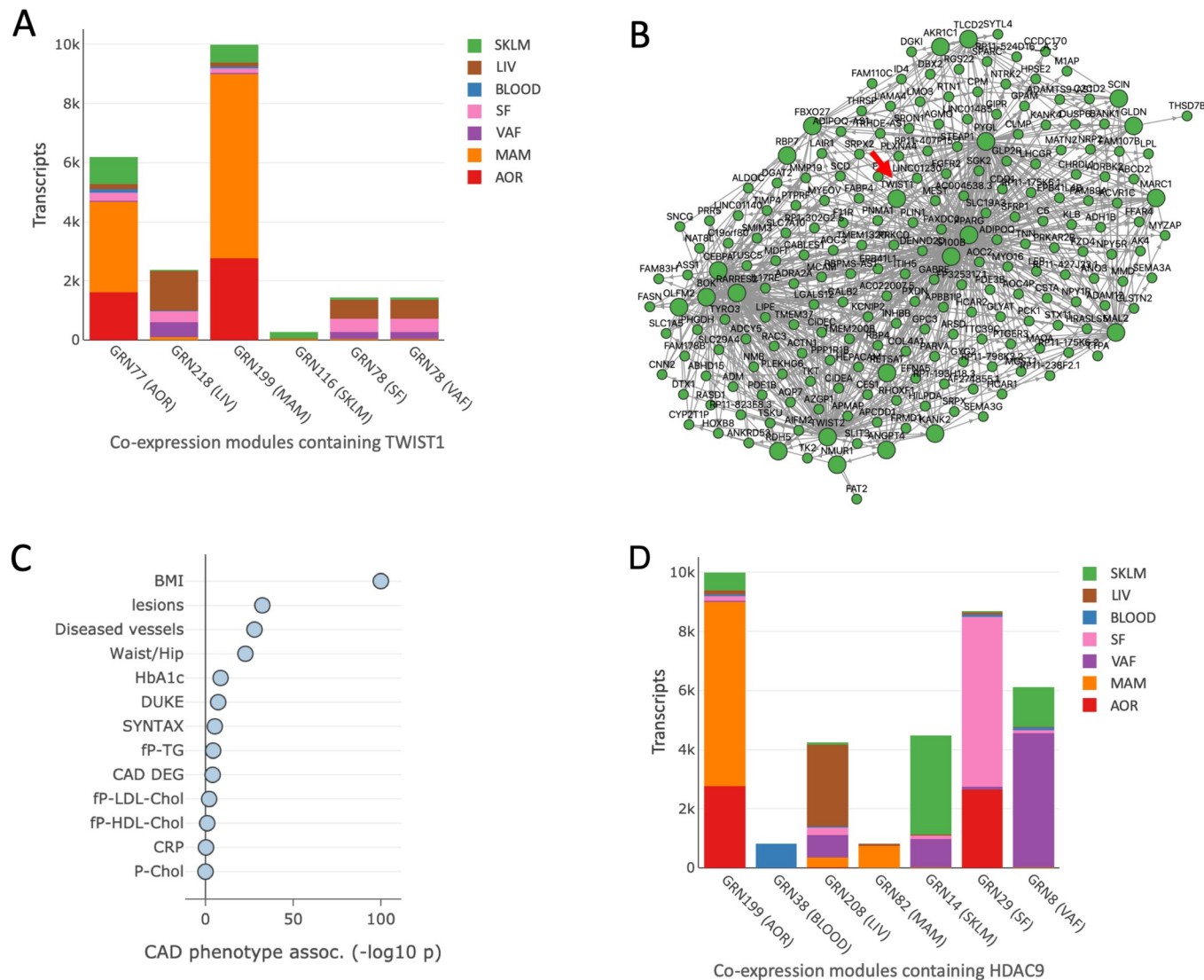

**Fig 7. GRNs involving *TWIST1* or *HDAC9* and their associations in STARNET tissues.** GRNs were generated from AOR, MAM, BLOOD, LIV, SKLM, SF and VAF tissues in STARNET as described [10]. **(A)** Of a total of 224 GRNs identified across these STARNET tissues, *TWIST1* was found in 5 (GRNs 77, 78, 116, 199, 218), where it acted in AOR, MAM, LIV, SKLM, SF and VAF (*TWIST1* acted in GRN 78 in both SF and VAF). Color codes represent differing tissues acquired in the STARNET study, while the tissue named on the horizontal axis (i.e. GRN 77 (AOR)) indicates the tissue containing *TWIST1* in that particular GRN. The vertical axis ('Transcripts') represents the total number of transcripts in each module. **(B)** Visualization of GRN 116, with *TWIST1* identified as hierarchically being among the top key drivers as indicated by red arrow (P = 1.63 x 10$^{-32}$). This GRN contains 279 genes, with 77.1% being in SKLM (including *TWIST1*), 15.8% in MAM, 3.9% in AOR and a smaller proportion in SF, BLOOD and LIV. Apart from GRN 116, *TWIST1* was not a key driver in any other GRNs. **(C)** Association of GRN 116 with clinical traits assessed in the STARNET study [10]. CAD DGE represents the enrichment of differential gene expression in the module between cases and controls. GRN 116 was associated with certain metabolic indices and also with the severity of CAD as assessed by number of coronary lesions ("lesions"), number of diseased vessels ("Diseased vessels"), SYNTAX score ("SYNTAX") and Duke CAD index. **(D)** GRNs involving *HDAC9* in STARNET. HDAC9 was found in 7 different GRNs (GRNs 8, 14, 29, 38, 82, 199, 208), where it acted in AOR, MAM, LIV, SKLM, BLOOD SF and VAF, but it was not a key driver in any of these networks. These data are also available from the dedicated STARNET website described in Koplev et al (starnet.mssm.edu) [10].

objective tool for understanding the causal mechanisms of disease-associated SNPs and their genetic loci. Our overarching finding is that the disease-causal effects of the *HDAC9*-associated CAD risk locus are governed via *TWIST1*, which reconciles conflicting prior studies [5,12,16–22] and delineates the pathway forward for research related to this locus.

In detail, the major findings to emerge from this study include: 1) Despite that fact that a significant proportion of the SNPs in the *HDAC9*-associated risk locus physically reside within the *HDAC9* gene (Figs 1 and 3), the effects of this locus are mediated through modulation of *TWIST1* expression levels in the arterial wall; 2) The lead SNPs rs2107595 also exhibited the strongest eQTLs in the arterial wall (S1 Table), adding to the likelihood that it is the true CAD-associated SNP in this locus; 3) The risk alleles were consistently associated with increased *TWIST1* levels, which combined with the MR analysis, indicate that increased arterial *TWIST1* levels are causal for atherosclerosis and CAD; 4) Although prior studies suggested that this locus may cause atherosclerosis and CAD via HDAC9 signaling in inflammatory cells [19,20], and despite the fact that STARNET included macrophages (MP) and foam cells (FC), there were no eQTLs for *HDAC9* in any examined tissues in either STARNET or GTEx (S1 and S2 Tables); 5) The lack of any significant eQTLs for *HDAC9* cannot be explained by low gene expression levels, as these were comparable between *HDAC9* and *TWIST1* in all tissues (Figs 4 and S1); 6) SNPs at the *HDAC9*-associated risk locus do not impact HDAC9 protein structure or function (Fig 3).

As an important aspect of these findings, it should be highlighted that there was independent, internal validation of key results. First, our pipeline indicated a major and concordant relationship between increased *TWIST1* levels in the arterial wall and CAD. This was internally validated by the following results: A) The strongest eQTLs were for *TWIST1* in the arterial wall, with the risk alleles consistently being associated with increased *TWIST1* levels (Figs 5 and S2); B) In the MR analysis *TWIST1* was found to be causal for CAD in the arterial wall, again and as expected based on the eQTL results with increased levels of *TWIST1* found to promote CAD; C) *TWIST1* levels in the aortic wall were associated with burden of CAD (Duke CAD index), again with the same direction of association (increasing *TWIST1* levels were associated with increasing Duke CAD index). Of note, the positive association between *TWIST1* levels and Duke CAD index (C), was an entirely independent finding with respect to (A) and (B). Finally, we leveraged an orthogonal analysis of single-cell chromatin accessibility profiles in human coronary artery to identify a significant linkage between candidate regulatory elements overlapping the lead CAD GWAS SNP near *HDAC9* (rs2017595) and the *TWIST1* promoter. These internal, independent validations of these core findings relating to *TWIST1* in STARNET and coronary artery tissues, collectively underscore the rigor and reproducibility of both our informatics pipeline and the results of this study.

TWIST1 is a basic helix-loop-helix transcription factor that is known to govern gene expression and multiple biological processes such as endothelial-to-mesenchymal transition [50,51], while also playing an important role in developmental processes and various disease states [52,53] including atherosclerosis [54]. Future studies to map the DNA binding sites of TWIST1 in different cellular and disease contexts will help resolve its precise transcriptional effects on candidate genes. Nevertheless, we speculate that TWIST1 governs the expression levels of a large number of genes, and that by modulating gene expression levels in the arterial wall TWIST1 plays a key role in the development and progression of atherosclerosis and CAD. In addition, *TWIST1* was found to be a key driver of GRN 116 in SKLM. In the context of atherosclerosis and CAD, SKLM is considered to play a role in metabolic functioning–a fact that was borne out by the most prominent phenotypic association of GRN 116 being BMI (Fig 7C). These findings are consistent with prior reports suggesting that levels of *Twist1* in adipocytes are related to mitochondrial function, insulin resistance and BMI [55,56].

While our study showed that the *HDAC9*-associated CAD risk locus does not cause CAD by modulation of *HDAC9* levels, structure or function, our study does not suggest that HDAC9 is unimportant in causing atherosclerosis and CAD. On the contrary, this study showed that *HDAC9* levels in AOR were positively associated with the burden of CAD (S3

Table). Consistent with this, our group just completed a separate study showing the importance of Hdac9 in a mouse model of atherosclerosis, whereby knockout of *Hdac9* in endothelial cells was associated with reduced burden of atherosclerosis [18]. Highlighting the value of our data-driven informatics pipeline [7], our original reason for embarking on this analysis of the *HDAC9*-associated CAD risk locus was to understand its role in governing the CAD- and atherosclerosis-causal effects of HDAC9. Due to the results presented here we now have a clear understanding that the *HDAC9*-associated CAD risk locus is not involved in the CAD-causal effect of HDAC9. This has allowed us to avoid wasting time and resources conducting laborious gene editing or other related experiments seeking to change the alleles at the SNPs in this locus for the purpose of modulating HDAC9 levels or function. Rather, because of the results presented here we have channeled our ongoing efforts into exploring other means to manipulate HDAC9 for therapeutic gain.

There are certain limitations of this study. First, since its description in 2020 [7], we have continued to improve this pipeline and it is likely that future refinements will further increase its efficiency and accuracy. In addition, although we describe this approach as a 'pipeline' (which typically implies seamless end-to-end data processing with minimal user supervision), significant human interpretation of the results is required. It is hoped that this 'pipeline' approach to understand key risk loci can become less reliant on human interpretation as it evolves. Second, we used STARNET as our main transcriptomic dataset. STARNET was collected on CAD patients and while GTEx can be applied to a broader range of individuals, it is not disease specific. Therefore, the ability to apply our pipeline to other diseases may be dependent on the availability of appropriate disease-specific transcriptomic datasets. Furthermore, both STARNET and GTEx used bulk (whole tissue) RNA sequencing, and did not use single cell RNA sequencing. While we queried a recent single-cell ATAC-seq dataset in coronary artery, future efforts to create large-scale CAD-relevant single cell multi-omic datasets will further improve the precision of this pipeline to refine disease-causal effects in specific cell populations and transition states, rather than just specific tissue types.

In conclusion, we applied our data-driven informatics pipeline [7] to gain mechanistic insights on the *HDAC9*-associated CAD risk locus. The pipeline efficiently and successfully revealed the disease-causal effects of this locus, and via multiple consistent lines of evidence, showed that its CAD-causal effects are primarily driven by modulation of *TWIST1* expression levels in the arterial wall. These results should inform future research efforts to experimentally decipher the *HDAC9*-associated CAD risk locus. From the broader perspective, our pipeline provides a powerful and efficient informatic strategy to understand the complex nature of genetic risk loci associated with common diseases, prior to embarking on their experimental validation.

## Supporting information

**S1 Fig. Gene expression levels in GTEx. (A)** Gene expression levels of *HDAC9*, *TWIST1*, *FERD3L* and AC003986.6 in GTEx–generally showing higher expression levels of *HDAC9* and *TWIST1* across various tissue subtypes. **(B)** Same as A but at different scales for each gene. RPKM, Reads Per Kilobase of transcript per Million mapped reads.
(TIF)

**S2 Fig. Cis-eQTLs for *TWIST1* in the *HDAC9*-associated CAD risk locus in AOR in GTEx.** eQTL violin plots are shown for tissue expression of *TWIST1* in AOR in GTEx corresponding to SNPs in the *HDAC9*-associated CAD risk locus. The CAD risk allele is presented on the right in all panels, indicating that the CAD risk allele is consistently associated with higher expression levels of *TWIST1* in AOR. There are no FDR or adjusted P value considerations

made in GTEx (this is not possible through the GTEx website). This Figure corresponds to S2 Table.
(TIF)

**S3 Fig. UMAP (Uniform Manifold Approximation and Projection) feature plot showing the ArchR-based imputed gene score activity for *TWIST1* derived from coronary artery scATAC-seq data. (A)** UMAP plot showing the greater chromatin accessibility in smooth muscle cells (SMC) and fibroblasts relative to other cell types for *TWIST1*. **(B)** UMAP feature plot from integrated scATAC/scRNA-seq data in coronary artery, which more clearly shows *TWIST1* expression in modulated SMCs annotated as fibromyocytes. Labelling of annotated clusters is based on top defining markers from Turner et al. [45].
(TIF)

**S4 Fig. Association of GRNs involving *HDAC9* with clinical traits assessed in the STAR-NET study.** Shown here are the 7 GRNs that involve *HDAC9* (GRNs 8, 14, 29, 38, 82, 199, 208) and their associations with clinical traits in the STARNET study. The tissue in which *HDAC9* is a member of each of these GRNs is shown in brackets (e.g. 199 (AOR), signifies that *HDAC9* expressed in the Aorta participates in GRN 199). With the exception of GRN 208, each of these was associated with the severity of CAD as assessed by number of coronary lesions ("Lesions"), number of diseased vessels ("Diseased vessels"), SYNTAX score ("SYNTAX") or Duke CAD index ("DUKE"). These data are also available from the dedicated STARNET website described in Koplev et al (starnet.mssm.edu) [10].
(TIF)

**S1 Table. Cis-eQTLs in the *HDAC9*-associated CAD risk locus in STARNET.** This analysis was performed using STARNET datasets, in all 9 STARNET tissues/cells (SF, VF, AOR, LIV, SKLM, BLOOD, MAM, MP, FC). eQTLs are shown for FDR P $\leq$ 0.05. Note that this analysis of cis-eQTLs sought and identified all eQTLs within 1 MB upstream or downstream from the genes in the *HDAC9*-associated CAD risk locus (i.e. that were in cis), which included the genomic regions encoding for *HDAC9*, AC003986.6, *FERD3L* and *TWIST1*. There were no cis-eQTLs identified for *HDAC9* or *FERD3L*.
(XLSX)

**S2 Table. Cis-eQTLs in the *HDAC9*-associated CAD risk locus in GTEx.** This analysis was performed using GTEx datasets, in the GTEx tissues SF, VF, AOR, LIV, SKLM, BLOOD, TA and COR. GTEx only provides P values, therefore there are no FDR or adjusted P value considerations made in this table (this is not possible through the GTEx website). As per S1 Table, this analysis of cis-eQTLs sought and identified all eQTLs within 1 MB upstream or downstream from the genes in the *HDAC9*-associated CAD risk locus (i.e. that were in cis), which included the genomic regions encoding *HDAC9*, AC003986.6, *FERD3L* and *TWIST1*. There were no ciseQTLs identified for *HDAC9* or *FERD3L*.
(XLSX)

**S3 Table. Phenotype correlations of *TWIST1* and *HDAC9* expression levels in AOR.**
*TWIST1* levels in AOR were inversely associated with body mass index (BMI) and HbA1c levels, and positively associated with Duke CAD index. There were also non-significant trends for other markers of CAD severity (SYNTAX score and the number of diseased coronary vessels). *HDAC9* levels in AOR were positively associated with the severity of CAD as assessed by number of diseased coronary vessels and with a trend also for Duke CAD index.
(XLSX)

## Author Contributions

**Conceptualization:** Yang Xu, Ke Hao, Clint L. Miller, Jason C. Kovacic.

**Data curation:** Lijiang Ma, Nicole S. Bryce, Adam W. Turner, Antonio F. Di Narzo, Karishma Rahman, Raili Ermel, Katyayani Sukhavasi, Oscar Franzen, Clint L. Miller.

**Formal analysis:** Lijiang Ma, Nicole S. Bryce, Adam W. Turner, Antonio F. Di Narzo, Ke Hao, Clint L. Miller, Jason C. Kovacic.

**Funding acquisition:** Johan L. M. Björkegren, Jason C. Kovacic.

**Investigation:** Lijiang Ma, Yang Xu, Raili Ermel, Katyayani Sukhavasi, Valentina d'Escamard, Nirupama Chandel, Bhargavi V'Gangula, Kathryn Wolhuter, Daniella Kadian-Dodov, Oscar Franzen, Arno Ruusalepp, Ke Hao, Johan L. M. Björkegren, Jason C. Kovacic.

**Methodology:** Lijiang Ma, Nirupama Chandel, Bhargavi V'Gangula, Kathryn Wolhuter, Daniella Kadian-Dodov, Oscar Franzen, Arno Ruusalepp, Ke Hao, Johan L. M. Björkegren, Jason C. Kovacic.

**Project administration:** Raili Ermel, Valentina d'Escamard, Arno Ruusalepp, Johan L. M. Björkegren, Jason C. Kovacic.

**Resources:** Raili Ermel, Katyayani Sukhavasi.

**Software:** Oscar Franzen.

**Supervision:** Ke Hao, Johan L. M. Björkegren, Jason C. Kovacic.

**Visualization:** Nicole S. Bryce, Clint L. Miller, Jason C. Kovacic.

**Writing – original draft:** Lijiang Ma, Antonio F. Di Narzo, Karishma Rahman, Yang Xu, Jason C. Kovacic.

**Writing – review & editing:** Nicole S. Bryce, Adam W. Turner, Antonio F. Di Narzo, Karishma Rahman, Yang Xu, Raili Ermel, Katyayani Sukhavasi, Valentina d'Escamard, Nirupama Chandel, Bhargavi V'Gangula, Kathryn Wolhuter, Daniella Kadian-Dodov, Arno Ruusalepp, Ke Hao, Clint L. Miller, Johan L. M. Björkegren, Jason C. Kovacic.

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
