## [Decision Letter · Decision Letter 0]

12 Jan 2022

Dear Dr Kovacic,

Thank you very much for submitting your Research Article entitled 'An Efficient Informatics Pipeline to Understand Complex Gene Loci: The HDAC9-associated risk locus promotes coronary artery disease by governing TWIST1' to PLOS Genetics.

The manuscript was fully evaluated at the editorial level and by independent peer reviewers. The reviewers appreciated the attention to an important problem, but raised some substantial concerns about the current manuscript. Based on the reviews, we will not be able to accept this version of the manuscript, but we would be willing to review a much-revised version. We cannot, of course, promise publication at that time.

If you decide to revise the manuscript for further consideration at PLOS Genetics, please aim to resubmit within the next 60 days, unless it will take extra time to address the concerns of the reviewers, in which case we would appreciate an expected resubmission date by email to plosgenetics@plos.org.

[LINK]

We are sorry that we cannot be more positive about your manuscript at this stage. Please do not hesitate to contact us if you have any concerns or questions.

Yours sincerely,

Marie Guerraty

Guest Editor

PLOS Genetics

Gregory Barsh

Editor-in-Chief

PLOS Genetics

Reviewer's Responses to Questions

**Comments to the Authors:**

Reviewer #1: In this manuscript, Ma and colleagues present their analysis of the causal variation and gene responsible for the CAD association at the 7p21.1 locus. This is an important question, as this locus is highly associated with multiple vascular diseases, and there is controversy regarding the causal gene at this locus, with some studies pointing to HDAC9 and others to TWIST1. The authors embark on a comprehensive suite of analyses to identify the most likely causal SNP and the causal gene, which they conclude to be the lead SNP rs2107595 and TWIST1, respectively. The analyses of the locus architecture, and the eQTL-based analyses of gene causality are solid. In particular, the human single cell ATAC data contributes new, complementary evidence that strengthens the linkage between rs2107595 and TWIST1. The study also fails to replicate the previously reported eQTL for HDAC9 in inflammatory cells, which is potentially noteworthy. However, other analyses (eQTL, Mendelian Randomization) have been previously done for this locus, also with using STARNET data, and have reached similar conclusions. There are also significant issues with the TWIST1 correlation and GRN analyses that limit their interpretability. However, this study does provide important incremental evidence to support the assertion that TWIST1 is the causal gene at this important locus.

MAJOR COMMENTS:

A very similar mendelian randomization analysis has previously been performed for TWIST1 and CAD, with a significant MR association between TWIST1 and CAD using the same resource (STARNET mammary artery/internal thoracic artery, https://doi.org/10.1016/j.ajhg.2017.04.016, supplemental table 3), rendering the current eQTL-based analyses largely redundant/confirmatory.

Regarding the correlation of TWIST1 expression levels to other genes in the artery wall, the finding that 16,472 genes were ‘significantly’ associated with TWIST1 expression despite Bonferroni correction is difficult to fathom, given there are only ~20,000-25,000 genes in the human genome. In addition, there is a strong chance that many of these associations are simply driven by cell type - i.e. if TWIST1 is expressed highly in a cell type within the bulk RNAseq data, all other genes in that cell type will be highly correlated only by virtue of their elevated expression in that cell type, not because they are in a pathway with TWIST1.

Fig. 7G - the GO processes presented here are extremely generic; it is difficult to ascribe any specific role to TWIST1 using these GO terms. This may be related to including too many ‘significant’ gene associations as per the above comment.

The authors infer that the presence of HDAC9 in 7 GRNs ‘confirms’ that HDAC9 is likely involved in the pathogenesis of CAD - to support this statement, we would need to know whether the GRNs in which HDAC9 resides are also associated with CAD phenotypes, similar to the TWIST1-containing GRNs.

Fig. 8 - the use of multiple different tissues to form GRNs is problematic in my mind, for a similar reason that I object to correlation of TWIST1 with other transcripts from bulk tissue samples containing multiple cell types. For example, gene expression will vary dramatically between tissues, causing strong correlations in the GRNs, but most often these genes do not actually have a real regulatory relationship and instead are merely correlated across tissues.

MINOR COMMENTS:

I am not sure that a ‘pipeline’ is the best way to describe the set of analyses in this manuscript. A pipeline often refers to processing of data in an end-to-end fashion with minimal user supervision outside of the adjustment of parameters. This manuscript describes a collection of discrete analyses, each requiring intensive human interpretation.

LD scores of the GWAS SNPs were calculated using 1000 Genome data - in what ancestry group(s)?

Regarding GTEx tissues - why not also use tibial artery? Expected to have many common features with AOR/COR and may increase power

Given similar TWIST1 and HDAC9 expression in arterial tissues, the authors may want to explicitly state that the lack of a significant HDAC9 eQTL in these tissues cannot be explained by low HDAC9 levels.

Regarding the lack of a significant eQTL relationship for HDAC9 in macrophages and foam cells, how many samples for each are present in the STARNET data, and what were the levels of HDAC9 in these samples?

Fig.6 - please include a UMAP of cell clusters from the scATAC data or scATAC/scRNA combined data in this figure. Top defining markers of each cluster should also be given.

Fig. 6 - what is the ‘coronary’ track? Is this a composite of all cell types?

Fig. 6 - From this view, there does not appear to be a great deal of open chromatin at the putative enhancer sites harboring rs2107595 and its LD SNPs. A zoomed in view of this region would be helpful to the reader. Also, given rs2107595 is the putative causal SNP, it would be helpful to see this SNP annotated in a different color vs the other SNPs.

Fig. 6 - when calculating the peak to gene correlation (rs2107595-containing peak vs TWIST1 expression), was the correlation calculation performed within each cell type or across all cell types?

Regarding the association of TWIST1 expression levels with CAD severity and other CAD-related phenotypes - the authors should compare this to a similar analyses with HDAC9 expression.

Fig. 8 - the authors state that key drivers cannot be determined for GRNs containing >3500 genes, and unfortunately the GRNs derived from the most relevant tissues (AOR and MAM) fit this criterion. Is there any way to modify the GRN parameters so that key drivers can be determined in the relevant tissues?

In the discussion, what is meant by TWIST1 exerting ‘additional’ effects by appearing in a GRN in skeletal muscle? What evidence is there that TWIST1 affects CAD via skeletal muscle, versus simply appearing as a key driver in a skeletal muscle network due to one of the many idiosyncrasies of GRN construction? If the authors wish to assert that TWIST1 affects CAD via skeletal muscle, they should back up this (somewhat unlikely) claim with experimental data.

Regarding the consistent directionality of the TWIST1 eQTL and MR results with respect to CAD (points A and B), given that both are derived from the same eQTL resource, these findings are expected to go in the same direction and are not independent validation of one another. Point ‘C’ is nice because it uses a different phenotype (Duke CAD index).

Reviewer #2: Ma et al, performed a deep fine mapping of the HDAC9 CAD risk loci and demonstrated, using a succession of bio-informatic tools that the potential causal gene is the region is TWIST1. Furthermore, the authors suggested that the mechanism of action of TWIST1 in CAD might be through the regulation of other genes rather than a direct transcriptional modification of the later. This work is particularly important in highlighting the need of using available data and several bio-informatic tools to explore candidate variants from GWAS before embarking on several experiments that explore the causality and biological implication of GWAS candidates’ loci. Below are my comments.

My main concern is about the Mendelian Randomization. Why did the authors choose the sum stat from Nelson et al rather than a much recent and powerfull summary statistic? Also, it is not clear what are the effect estimate used as instruments in the MR. if the lead SNP rs2107595 is significant at the GWAS level and significant for eQTL in AOR and MAM, then using eQTL from STARNET and the GWAS sum statistic violates the principle of MR.

The authors need to provide the effect estimate of the CAD GWAS and eQTL used for the MR

It will be interesting to also have more insight on the functional prediction of associated SNPs in the locus

What was the rational in selecting only the lead SNPs for further analysis?

Titles “ An Efficient Informatics Pipeline to Understand Complex Gene Loci: The HDAC9- associated risk locus promotes coronary artery disease by governing TWIST1” read as the authors are presenting a pipeline that they develop in the current paper however, it seem like the pipeline have already been published and this work is just a continuation. The title should be modified accordingly.

Table 1: the authors need to clarify the source of the effect size in the table 1 and to which diseases (I suppose CAD) the effect size corresponds given that some SNPs have been reported in multiple phenotypes.

Figure 2: The authors should create a full LDheatmap (R2 between 0 and 1 for all SNPS include in the plot) that display the correlation for all SNPs; the current map display only R2>0.4 or R2>0.8 this is misleading as it doesn’t clearly show the full picture of LD between SNPs. WE expect to see a single block as the authors suggested.

ST1and ST2 should also have the results for HER1 and HDAC9 for comparison

ST5 has a typos in the title “Supplementary Table5. Phenotypic associatinos of TWIST1 expression levels in AOR in STARNET subjects”

**Have all data underlying the figures and results presented in the manuscript been provided?**

Reviewer #1: Yes

Reviewer #2: **No: **All data are not provide but which is expected as most of the eQTL, RNAseq are publicly available

PLOS authors have the option to publish the peer review history of their article (what does this mean?). If published, this will include your full peer review and any attached files.

Reviewer #1: No

Reviewer #2: **Yes: **Catherine Tcheandjieu

---

## [Editor Report · Decision Letter 1]

17 May 2022

Dear Dr Kovacic,

We are pleased to inform you that your manuscript entitled "The HDAC9-associated risk locus promotes coronary artery disease by governing TWIST1" has been editorially accepted for publication in PLOS Genetics. Congratulations!

Yours sincerely,

Marie Guerraty

Guest Editor

PLOS Genetics

Gregory Barsh

Editor-in-Chief

PLOS Genetics

**Data Deposition**

http://datadryad.org/submit?journalID=pgenetics&manu=PGENETICS-D-21-01392R1

**Press Queries**

---

## [Editor Report · Acceptance letter]

14 Jun 2022

PGENETICS-D-21-01392R1 

The HDAC9-associated risk locus promotes coronary artery disease by governing TWIST1 

Dear Dr Kovacic, 

We are pleased to inform you that your manuscript entitled "The HDAC9-associated risk locus promotes coronary artery disease by governing TWIST1" has been formally accepted for publication in PLOS Genetics! Your manuscript is now with our production department and you will be notified of the publication date in due course.

With kind regards,

Anita Estes

PLOS Genetics

On behalf of:
